# Comparative Therapeutic Exploitability of Acute Adaptation Mechanisms to Photon and Proton Irradiation in 3D Head and Neck Squamous Cell Carcinoma Cell Cultures

**DOI:** 10.3390/cancers13061190

**Published:** 2021-03-10

**Authors:** Annina Meerz, Sara Sofia Deville, Johannes Müller, Nils Cordes

**Affiliations:** 1OncoRay—National Center for Radiation Research in Oncology, Faculty of Medicine Carl Gustav Carus, Technische Universität Dresden, 01307 Dresden, Germany; annina.meerz@uniklinikum-dresden.de (A.M.); sarasofia.deville@onktherapeutics.com (S.S.D.); Johannes.Mueller@uniklinikum-dresden.de (J.M.); 2Helmholtz-Zentrum Dresden-Rossendorf (HZDR), Institute of Radiooncology—OncoRay, 01328 Dresden, Germany; 3German Cancer Consortium, Partner Site Dresden, German Cancer Research Center, 69120 Heidelberg, Germany; 4Department of Radiotherapy and Radiation Oncology, University Hospital Carl Gustav Carus, Technische Universität Dresden, 01307 Dresden, Germany

**Keywords:** HNSCC, photon irradiation, proton beam irradiation, 3D cell culture, molecular targeting

## Abstract

**Simple Summary:**

Treatment resistance is a major obstacle affecting the outcome of patients with head and neck squamous cell carcinomas (HNSCC). Proton beam therapy may be beneficial in the treatment of HNSCC due to optimized dose distribution and consequent sparing of healthy tissue. However, molecular data on tumor cell responses upon proton irradiation appear sparse. The aim of this study was to compare the acute adaptative kinome of HNSCC cell lines to photon and proton irradiation and elucidate their therapeutic potential. Despite pronounced differences in kinome profiles upon photon and proton irradiation, these differences failed to be therapeutically exploitable. Instead, our results reveal radiation type-independent sensitization upon pharmacological inhibition of selected targets.

**Abstract:**

For better tumor control, high-precision proton beam radiation therapy is currently being intensively discussed relative to conventional photon therapy. Here, we assumed that radiation type-specific molecular response profiles in more physiological 3D, matrix-based head and neck squamous cell carcinoma (HNSCC) cell cultures can be identified and therapeutically exploited. While proton irradiation revealed superimposable clonogenic survival and residual DNA double strand breaks (DSB) relative to photon irradiation, kinome profiles showed quantitative differences between both irradiation types. Pharmacological inhibition of a subset of radiation-induced kinases, predominantly belonging to the mitogen-activated protein kinase (MAPK) family, failed to sensitize HNSCC cells to either proton or photon irradiation. Likewise, inhibitors for ATM, DNA-PK and PARP did not discriminate between proton and photon irradiation but generally elicited a radiosensitization. Conclusively, our results suggest marginal cell line-specific differences in the radiosensitivity and DSB repair without a superiority of one radiation type over the other in 3D grown HNSCC cell cultures. Importantly, radiation-induced activity changes of cytoplasmic kinases induced during the first, acute phase of the cellular radiation response could neither be exploited for sensitization of HNSCC cells to photon nor proton irradiation.

## 1. Introduction

The management of head and neck squamous cell carcinomas (HNSCC) remains challenging reflected by the thirty years unchanged overall surviving rates of about 50% [1,2]. According to stage and disease subtype, surgery, chemotherapy, and radiotherapy are combined in a multimodal regimen [3,4]. Among various factors, resistance to radiochemotherapy and quality of life reducing normal tissue side effects lead to low tumor control rates, locoregional recurrence, and eventually treatment failure [5,6]. Consequently, optimized radiation beam delivery concepts and a better comprehension of the molecular resistance mechanisms commenced to be intensively evaluated [6].

Over the past years, our awareness of the potential of proton therapy for various tumor entities including HNSCC in clinical settings grew [7]. According to the well-defined energy deposition in depth, proton irradiation may have a favorable dosimetric profile consisting of a more optimal sparing of healthy tissues of adjacent organs at risk for achieving a decline in normal tissue toxicities [8,9,10]. This in turn may allow further dose escalation in the well-defined tumor area. Despite the undisputable differences in terms of physical characteristics, current data about the relative biological effectiveness of protons relative to photons remain inconsistent, hence, requiring further investigations to clarify benefits of proton beam therapy compared with conventional photon therapy [11,12].

With proton beam therapy presenting an excellent physical opportunity, our understanding of molecular resistance mechanisms is in its infancy. Multiple deregulated signaling networks controlling cellular processes such as survival, proliferation, and metastasis jointly operate in tumor cell resistance to therapy [13,14]. This also includes the DNA damage response induced by ionizing radiation [15]. At least preclinically, the targeting of DSB repair enzymes has been documented to represent a solid strategy to overcome intrinsic radioresistance in tumor cells [16]. While the body of literature for photon irradiation is large, comparable data for proton irradiation are sparse.

By means of validated three-dimensional (3D) cell cultures resembling in-vivo growth conditions [17], we comparatively explored clonogenicity, DSB repair, and kinome changes in HNSCC cultures exposed to either photon or proton irradiation. We hypothesized that radiation-induced changes in the kinome during the acute phase of the damage response are radiation type-specific and can therefore be therapeutically exploited to sensitize cells specifically to either photon or proton irradiation. We found cell survival and DSB repair of 3D HNSCC cells comparable upon photon and proton irradiation, while the acute kinome changes appeared largely different. Intriguingly, exploitation of these differences, particularly regarding signaling molecules of the mitogen-activated protein kinases (MAPK) axis was unsuccessful alternatively guiding us to test for the radiosensitizing potential of DNA repair inhibitors, which mediated radiosensitization undistinguishable between photon and proton irradiation.

## 2. Results

### 2.1. The Intrinsic Cellular Radiosensitivity to Photon and Proton Irradiation Varies among 3D lrECM HNSCC Cell Cultures

We commenced our study by evaluating the 3D clonogenic survival and found varying plating efficiencies and clonogenic radiation survival in our tested HNSCC cell line panel (Figure 1A–C). In four out of seven cell lines (UTSCC15, Cal33, UTSCC14, FaDu), a statistically significant difference between the radiation survival after photon versus proton irradiation was observed for single radiation doses (Figure 1C).

In addition, the analysis of residual γH2AX-positive foci with respect to number and size (Figure 1D–F) revealed no difference in the amount of foci upon photon or proton irradiation in four (UTSCC5, UTSCC15, UTSCC14, FaDu) out of seven cell lines (Figure 1E). In contrast, SAS and Cal33 cell cultures showed increased numbers of residual foci after proton relative to photon irradiation, while HSC4 cells, exclusively, displayed higher amounts of foci upon photon than proton irradiation. Foci size varied non-significantly in a cell line-dependent manner (Figure 1F). Of note, a statistically significant correlation between the clonogenic survival at 4 Gy and the amount of residual γH2AX-positive foci was observed merely after photon irradiation (Appendix A). Collectively, our data suggest a cell line-specific radiosensitivity and a repair of radiation-induced DSB that is independent from photon or proton irradiation.

### 2.2. Photon and Proton Irradiation Induce Differential Changes in Kinome Signatures

Next, we comparatively explored acute radiation-induced alterations in tyrosine and serine/threonine kinomes at 2 h after irradiation. For this purpose, SAS and UTSCC15 cell cultures were intentionally chosen as representative models for the following two reasons: (i) similarity of their intrinsic radiation sensitivity to protons and photons, (ii) lack of clear radiation-type-specific differences in clonogenic survival in 6 out of 7 HNSCC cell lines. Overall, we detected obvious changes in activation patterns of serine/threonine (STK) (Figure 2A,E) and phosphotyrosine kinases (PTK) (Figure 2B,F) in both cell lines upon 4 Gy of either photon or proton irradiation. The visual evaluation of the heatmaps indicates an overall downregulation of kinase activities in SAS after photon irradiation and even stronger after proton irradiation in the panel of tested kinases, whereas a few increased kinase activities became observable for UTSCC15. Generally, SAS cells revealed stronger kinase activity changes than UTSCC15 cells (Figure 2C,G). In Figure 2D,H, we demonstrate that SAS cells showed a higher number of downregulated STK and PTK relative to UTSCC15 cells after photon and proton irradiation. Intriguingly, UTSCC15 but not SAS cells revealed a general radiation-induced upregulation of STK kinase activity when exposed to 4-Gy proton irradiation (Figure 2B,C). To comply with our translational intent to target radiation-hyperactivated kinases for sensitization, we subsequently focused on serine/threonine kinases.

### 2.3. Proton Irradiation Predominantly Induces Mitogen-Activated Protein Kinases

By calculating the ratio between mean STK activities upon proton irradiation and photon irradiation, the difference in STK activation patterns between irradiated 3D SAS and UTSCC15 cell cultures became more obvious showing that all STK, except DCAMKL1, are induced in UTSCC15 cells, while almost all STK are downregulated in SAS cells (Figure 3A). 

Plotting these data in Figure 3B displays 66 kinases less active in SAS cells and more active in UTSCC15 cells upon proton versus photon irradiation in the left upper quadrant. Intriguingly, several kinases belonging to different mitogen-activated protein kinase signaling cascades clustered in this left upper quadrant (Figure 3B). For a more rigorous analysis with an expected higher biological relevance, we defined a downregulation cut-off of ≤−0.2 in SAS cells and an upregulation cut-off of ≥0.2 in UTSCC15 cells for the 66 kinases selected in Figure 3B. Consequently, 35 kinases were eligible to undergo pathway analysis by Reactome confirming an involvement of more than 80% of these analyzed kinases in MAPK activation pathways and/or nuclear events mediated by MAPK (Figure 3C). By specifically summarizing the activities of these kinases, it became obvious that SAS and UTSCC15 cells show opposing activation patterns (Figure 3D and Appendix A).

To further underpin the impact of MAPK associated molecules for the survival of patients with HNSCC, we analyzed correlated mRNA levels of the selected MAPK kinases from our kinome with survival data from 515 HNSCC patients from the TCGA (PanCancer Atlas). Importantly, the connection of MAPK associated signaling molecules and their potential to radiosensitize are well documented for photon irradiation [18,19]. However, to our knowledge there is no dataset for proton irradiation. The relation between activation patterns and Spearman’s correlation analyses was performed among the 11 identified MAPK kinases from the kinome dataset. While the various tested combinations of MAPK associated kinases indicated mostly highly significant correlation with overall survival Appendix A), the mRNA analyses revealed an inconsistent expression pattern depending from kinase to kinase, reflecting the identified differences between SAS and UTSCC15 cell lines (Appendix A). Interestingly, MAPK11 (p38β) and MAPK12 (p38γ) were significantly correlated, likewise, two other kinases from the p38 family, MAPK13 (p38δ) and MAPK14 (p38α) were also significantly and positively correlated (Appendix A). Nevertheless, no clear trend or clustering among the selected kinases was observable.

The general notion of varying MAPK regulation underpins the fact that understanding the complexity of MAPK signaling regulation may foster the elucidation of potential target molecules for HNSCC patients. Taken together, both radiation modalities lead to distinguishable patterns in kinase activity, especially for kinases involved in the MAPK signaling pathways.

### 2.4. Inhibition of the MAPK Signaling Molecules ERK1/2, p38 and JNK1/2/3 Is Not Specific for Clonogenic Survival upon Photon and Proton Irradiation

Next, we carried on by focusing on three of the key players of MAPK pathways (ERK1/2, JNK 1/2/3, p38α/β/ɣ/δ) (Appendix A). To elucidate whether these three MAPK enzymes play a critical role for clonogenic survival upon photon or proton irradiation, we applied selective inhibitors (Ulixertinib for ERK1/2; SP600125 for JNK 1/2/3; Ralimetinib for p38α/β/ɣ/δ). While basal clonogenic 3D HNSCC cell survival remained unaffected (SAS, UTSCC14) or only slightly reduced (UTSCC15, FaDu) by p38 and JNK inhibitors (Figure 4A), a decrease in plating efficiency was observed upon ERK inhibition in all cell lines ranging between 33% (FaDu) and 81% (UTSCC14) relative to DMSO (Figure 4B and Appendix A). 

Upon irradiation, we exhibited radiosensitization through pre-treatment with Ulixertinib compared with DMSO, while SP600125 and Ralimetinib pre-treatment generally failed in three out of four cell lines to mediate a radiosensitizing effect (Figure 4B and Appendix A). Intriguingly, we detected no difference in clonogenic survival between photon and proton irradiation in the tested 3D cell cultures, although MAPK activities were induced by proton irradiation in UTSCC15 and reduced in SAS cell cultures (Figure 4B). Additionally and based on the radiation-induced kinase activation, we conducted an approach consisting of inhibitor treatment 30 min post irradiation. Of note, similar results were obtained as observed for the inhibitor pre-treatment regimen (Appendix A).

As ERK inhibition emerged to hold the most promising radiosensitizing potential and is per se anti-proliferative, we evaluated, in addition to clonogenic survival, cell proliferation by measuring the size of 3D UTSCC15 and SAS cell colonies (Figure 4C,D). An overall and significant reduction in colony size was observed in both cell lines in presence of Ulixertinib relative to DMSO. Upon photon or proton irradiation, merely SAS demonstrated significantly reduced colony sizes (Figure 4D). Combined with irradiation, the colony size of ERKi-pretreated cell cultures did not significantly differ from that of ERKi mono-treatment (Figure 4D). Conclusively, differential ERK activities were observed after proton as compared to photon irradiation in UTSCC15 and SAS cell lines. However, the therapeutic exploitation of this difference using the ERK1/2 inhibitor could not be achieved as Ulixertinib was found to effectively sensitize both HNSCC cell lines in a radiation type- and schedule-independent manner. Interesting in this context is an analysis of TCGA data sets from patients with HNSCC, which corroborated ERK inhibition as promising approach [20] (Appendix A). In Appendix A, the mRNA expression levels (Appendix A) of selected MAPK associated signaling enzymes were combined for Kaplan-Meyer analysis showing a clear discrimination of survival curves for low versus highly expressed MAPK molecules (Appendix A). The apparent impact of these molecules on patient’s survival not only emphasizes its clinical relevance but simultaneously justifies a thorough investigation of the therapeutic potential of MAPK inhibitors combined with proton therapy.

### 2.5. Inhibitors for DNA Repair Proteins Elicited Similar Radiosensitization to Photon and Proton Irradiation

As alternative approach and based on our knowledge of the radiosensitizing potential of DNA repair inhibitors [15,21,22], we tested whether such inhibitors result in comparable radiosensitization of 3D HNSCC cell cultures exposed to photon or proton irradiation. Our observations in the ATM-deficient HNSCC cell line SKX, which showed higher radiosensitivity to protons relative to photons, prompted us to evaluate the efficacy of pharmacological inhibitors targeting a number of key enzymes of the two major DSB repair pathways non-homologous end joining (NHEJ) and homologous recombination (HR) [23]. We first discovered a high correlation between ATM expression levels, in contrast to all other investigated proteins (Figure 5C, Appendix A), and the enhancement ratios (calculated by dividing the surviving fraction at 4-Gy proton irradiation by the surviving fraction at 4-Gy photon irradiation) (Figure 5D). Of note, UTSCC5 cells seem to have an undescribed Rad51 deficiency.

Regarding basal 3D clonogenic survival, HNSCC cell cultures responded in a differential manner to the tested inhibitors (Figure 5E and Appendix A). In Figure 5F,G, it becomes clear that the examined inhibitor-pretreated cell lines do not largely discriminate between proton and photon irradiation with respect to clonogenic survival. Moreover, pharmacological inhibitors for DNA-PK, ATM and PARP achieved radiosensitization in a cell line-dependent manner in contrast to inhibitors for MRN, Rad51 and MDM2 (Figure 5F,G and Appendix A). In summary, our data corroborate the notion that a pharmacological targeting of DNA repair enzymes elicits radiosensitization of HNSCC cells. Importantly, the radiosensitizing potential seems to be dependent on which DNA repair protein is targeted but independent from the radiation type when comparing photon and proton irradiation.

## 3. Discussion

For the treatment of HNSCC, accumulating data has led to the assumption of a superiority of proton beam irradiation over photon irradiation due to higher precision and, thus, advanced sparing of healthy tissue [11]. Convincing preclinical studies along with supportive clinical data sets are, however, missing. Especially preclinical studies comparing early and acute tumor cell responses to proton versus photon irradiation are lacking, which may profoundly foster the radiation type-specific implementation of molecular-targeted agents. This preclinical study was performed in a panel of 3D, more physiologically grown HNSCC cell cultures to contribute towards deeper understanding of this issue.

Firstly, we observed equal efficacy of protons and photons regarding the clonogenic survival and DSB induction and repair, which is in line with the notion of similar radiobiological effectiveness between photons and protons [7,24]. Such similarities of photon and proton irradiation have also been found for pancreatic ductal adenocarcinoma [21], glioma stem cells [25] and lung cancer [26]. As a sufficient repair of DNA damage reflects one major determinant of intrinsic cellular radiosensitivity, the similarity in DSB repair in photon- and proton-irradiated cells further underscores our clonogenic survival data. Owing to a lack of gated stimulated emission depletion (STED) microscopy showing protons to induce more clustered and complex DNA damage [27], our analyses of foci number and size failed to show any great differences.

Following the idea of distinguishable biological responses between these two radiation types in presence of undistinguishable survival and DSB repair data, we hypothesized pronounced and therapeutically exploitable discrepancies between proton- and photon-related kinome profiles. The importance of this investigation lies in the fact that the only FDA approved targeted therapy for locoregionally advanced HNSCC is the anti-EGFR antibody Cetuximab, for which exists conflicting results about its efficacy [28]. Given the effort towards a personalized cancer therapy to overcome therapy resistances and treatment failure, the identification of prognostic biomarkers or novel cancer targets is of great interest [29]. Accordingly, the present study revealed i) differential patterns within serine/threonine and tyrosine kinase activities upon photon and proton irradiation and ii) a clustering of mitogen-activated protein kinase activity changes associated with a particular HNSCC cell line. In combination with bioinformatic TCGA data sets of HNSCC demonstrating stratification of patients into a high- and low-survivor group [20], this finding suggests radiation modality-dependent changes in MAPK pathway activity to play a role in the HNSCC cell radiation response. Although our limited approach to therapeutically exploit the kinome profiles failed, our results about ERK inhibition warrant further in-depth investigations and emphasize the rationale to include kinome analysis into our treatment decision making process. Considering that kinome profiling has been reported to be substantially valuable for investigating radioresistance in other tumor entities e.g., melanoma, pancreatic ductal adenocarcinoma or breast cancer [21,30,31], we were able to shed a light on kinase activity changes in HNSCC after proton therapy.

Moreover, future studies are warranted to investigate kinase activity changes in response to proton or photon exposure after different post-irradiation timeframes, as our investigation solely focused on early, acute adaptation mechanisms. As defined by us, this covers a limited timeframe of 2 h post-irradiation exposure.

As an alternative approach, targeting specific DSB repair proteins such as DNA-PK, ATM, and PARP efficiently reduced clonogenic survival in response to both radiation modalities but also failed to discriminate between photon and proton irradiation. Interestingly, published observations showed cell model-related dependencies on either HR or NHEJ. In monolayer A549 lung cancer cells and mouse embryonic fibroblasts, for example, HR was the predominant DSB repair pathway after proton irradiation [32,33]. In contrast, Gerelchuluum and colleagues published a NHEJ dependence in proton-irradiated Chinese hamster ovarian cells and lung fibroblasts [34]. Our findings indicate cell line-specific dependencies on HR and NHEJ regardless of the radiation type with preferences for DNA-PK, ATM, and PARP targeting. Obviously, there is not only very limited but also contradictory preclinical evidence of potential DNA damage pathway preferences.

Unquestionable at this point in time is that further molecular and in-vivo studies are necessary to verify our survival data as well as the differential responsiveness and radiosensitization seen for the panel of tested inhibitors.

## 4. Materials and Methods

### 4.1. Cell Culture

HNSCC cell lines (Cal33, FaDu, HSC4, SAS, UTSCC5, UTSCC14, and UTSCC15) were kindly provided by R. Grenman (Turku University Central Hospital, Turku, Finland). SKX cells were kindly provided by M. Krause (Technische Universität Dresden, Dresden, Germany). Cells were cultured in Dulbecco’s modified Eagle’s medium (DMEM, Sigma-Aldrich, Taufkirchen, Germany) supplied with 10% fetal calf serum (FCS, Sigma-Aldrich, Taufkirchen, Germany) and 1% non-essential amino acids (Sigma-Aldrich, Taufkirchen, Germany) at 37 °C in a humidified atmosphere containing 8.5% CO2 at pH 7.4. For all experiments asynchronously growing cells were used within passages 2–5 [21].

### 4.2. 3 D Colony Formation Assay

To assess colony formation ability, cells were seeded for 3D colony formation assay as described [35]. Briefly, two thousand cells were imbedded in 0.5 mg/mL laminin-rich extracellular matrix (lrECM (Matrigel™; BD Bioscience, Heidelberg, Germany)) and plated into 96-well-plates. Twenty-four h later, cells were treated with 10 µM of different inhibitors 1 h prior to 2, 4 or 6 Gy exposure of either photon or proton irradiation. After a cell line specific incubation time of 9–12 days, colonies (clusters of minimum 50 cells) were quantified under the microscope. The points on the surviving curves display the mean surviving fraction from independent experiments performed in trials.

### 4.3. Exposure to Photon and Proton Irradiation

Irradiation of cells was performed at room temperature with selected dose points of 2, 4, or 6 Gy of 200-kVp X-rays (Yxlon Y.TU 320; Yxlon; dose rate ≈ 1.3 Gy/min at 20 mA) filtered with 0.5-mm Cu, as published [35]. The absorbed dose was measured using a Semiflex ionization chamber (PTW Freiburg; Freiburg, Germany). Cells in tissue culture plates were irradiated horizontally both for colony formation assays (96-well plates) and whole-cell lysates (24-well plates) as published [35]. Proton irradiation (low-LET of 3.7 keV/µm) was generated by the cyclotron at the horizontal fixed-beam beam line in the experimental hall of the University Proton Therapy Dresden (UPTD) using a dedicated beam shaping system consisting of a double-scattering device and a ridge filter provides a laterally extended 10 × 10 cm^2^ proton field and a SOBP of 26.3 mm (90% dose plateau) in water to deliver 150 MeV protons. To assure mid-SOBP position, tissue culture plates were irradiated at room temperature using two different experimental settings. For colony formation assays 96-well plates were set up perpendicular to beam axis (90°), whereas for whole-cell lysates 24-well plates were placed at 42° relative to beam axis to circumvent damage of 3D structure [36]. To warrant quality insurance, a Markus ionization chamber (PTW) readout by an Unidos dosimeter (PTW) at sample position was applied for absolute dosimetry prior to irradiation. Further information of absolute dosimetry and beam control are specified elsewhere [37].

### 4.4. Foci Assay

1.5 × 10^6^ cells per well were seeded for γH2AX foci assay and cultured in lrECM (0.5 mg/mL) for 24 h. Cells were irradiated with 4 Gy of either photon or proton irradiation as published [38]. Twenty-four h post-irradiation, cells were harvested, fixed and permeabilized using 3% formaldehyde and Triton-X-100, respectively. Then, DSB were stained with anti-γH2AX (Ser139) antibodies and representative pictures were taken with Axioscope Z1 fluorescence microscope (Zeiss, Jena, Germany). Semi-automatic nucleus segmentation and the determination of foci number and size were done with a script written and implemented in Fiji [39]. In a first step, the DAPI-channel was thresholded with an iterative method. The segmented nuclei in the context of each nucleus’ γH2AX signal were then visually reviewed by the user in order to exclude mitotic or mal-segmented nuclei from further analysis. The number of foci in the γH2AX color channel were counted nucleus-wise using a maxima finder with a prominence value of 2000. Lastly, the position of the local maxima was used to obtain a foci-wise tessellation of the nucleus area. The size of an individual foci was then defined as the area with a grayvalue I above a threshold of IBackground + 0.5 (Imax—IBackground), which refers to a full-width half-maximum (FWHM) criterion. The area of each foci was measured separately for subsequent analysis. At least 100 cells per condition were evaluated manually to validate the script. Student’s *t*-test was performed with Prism7 GraphPad.

### 4.5. Antibodies

Antibodies for Western blotting were purchased as indicated: β-actin from Sigma-Aldrich (Taufkirchen, Germany) and GAPDH, DNA-PK, ATM, PARP, NBS1, ERK1/2 and phosphoERK1/2 from Cell Signaling (Frankfurt a.M., Germany). Antibody against Rad51 was from Invitrogen (Karlsruhe, Germany) and MDM2 from Santa Cruz (Heidelberg, Germany). γH2AX antibody was from Cell Signaling (Frankfurt a. M., Germany). Alexa Fluor 488 goat anti-rabbit IgG was from Invitrogen (Karlsruhe, Germany).

### 4.6. Inhibitors

Cells were treated with pharmacological inhibitors for ATM (KU55933, Calbiochem, San Diego, CA, USA), DNA-PK (NU7026, Selleckchem, Houston, TX, USA), MDM2 (AMG232, Axon Medchem, Groningen, The Netherlands), Rad51 (B02, Axon Medchem, Groningen, The Netherlands), MRNcomplex (Mirin, Sigma-Aldrich, Taufkirchen, Germany) and PARP (Olaparib, Cell Signaling, Frankfurt a. M, Germany). Inhibitors for ERK1/2 (Ulixertinib) and p38α/β/ɣ/δ (Ralimetinib) were obtained from Sellekchem (Houston, TX, USA). Inhibitor for JNK1/2/3 (SP600125) was purchased from Santa Cruz (Heidelberg, Germany). All the inhibitors were diluted with DMSO to a concentration of 10 mM and stored at −80 °C. For treatment, the inhibitors were diluted in DMEM supplied with 10% FCS and 1% non-essential amino acids at a concentration of 1 µM or 10 µM. DMSO was used as control for all inhibitors.

### 4.7. Total Protein Extracts and Western Blotting

Cell lysis was performed from cells irradiated with 0 Gy and 4 Gy of either photon or proton treatment and subsequently whole cell lysates were used for Western blotting as previously described [35]. After SDS-PAGE and protein transfer onto the nitrocellulose membrane, specific proteins were detected by using indicated primary antibodies and horseradish peroxidase-conjugated donkey anti-rabbit and sheep anti-mouse secondary antibodies. Chemiluminescent detection reagent was used for detection of proteins with a Western Blot Imager (Fusion FX, Vilber) and ImageJ for densitometry. In Appendix A, the original uncropped images of Western Blot are displayed.

### 4.8. Kinome Analysis

Kinase activity profiling with PamGene^®^ technology was performed using PamChip^®^ peptide microarray as published [40]. The kinase activities of 144 phosphotyrosine kinases and 140 serine/threonine kinases were determined in a PamStation 12 system according to the manufacturer’s instruction. For each condition, 1.5 × 106 cells in 0.5 mg/mL lrECM were seeded in 24-well plates prior coated with 1% agarose to avoid adhesion. Cells were irradiated with 4-Gy photon or proton irradiation after 24 h. The control was left unirradiated. Two h post-irradiation, cell lysis was performed in 3× kinase buffer (Cell Signaling, Frankfurt a. M., Germany), supplemented with HALT phosphatase and protease inhibitor cocktail (Thermo Scientific, Darmstadt, Germany). Then, snap-frozen supernatants of three-independent experiments were sent to the Genomics and Proteomics Core Facility Microarray Unit center at DKFZ for kinase activity analysis.

### 4.9. Data Analysis

The means ± standard deviation (SD) of at least three independent experiments (indicated as *n*) were calculated with reference to controls defined in total numbers or 1.0. For statistical significance analysis of clonogenic survival and densitometry two-sided Student’s *t*-test were performed using Microsoft Excel 2019 or Prism7. *p*-Value of less than 0.05 was considered statistically significant.

## 5. Conclusions

In summary, our preclinical study comparing acute adaptation mechanisms after photon and proton irradiation unravels superimposable, cell line-specific sensitivities and DNA double strand repair to photon and proton irradiation. Instead, kinome analysis revealed distinguishable kinase activity patterns after both irradiation types. Proton irradiation-induced kinases predominantly belong to the mitogen-activated protein kinase pathways, whose pharmacological inhibition resulted in radiosensitization but without a radiation type-related specificity. Likewise, targeting DNA repair enzymes—DNA-PK, ATM and PARP—elicited a radiosensitization without a clear higher efficacy of one radiation type over the other. Consequently, a more thorough examination is necessary for a better understanding of the molecular response patterns induced by proton and photon irradiation. For a translation to the clinic, future studies are warranted on other endpoints as well as on the underlying molecular mechanisms. Although this project implemented more physiological 3D, matrix-based cell cultures, patient-derived cultures and animal models would highly support the clarification of the potential of proton radiotherapy in HNSCC and, especially, of which molecular intervention can benefit patient’s survival as adjuvant to standard radio(chemo)therapy.

## Figures and Tables

**Figure 1 cancers-13-01190-f001:**
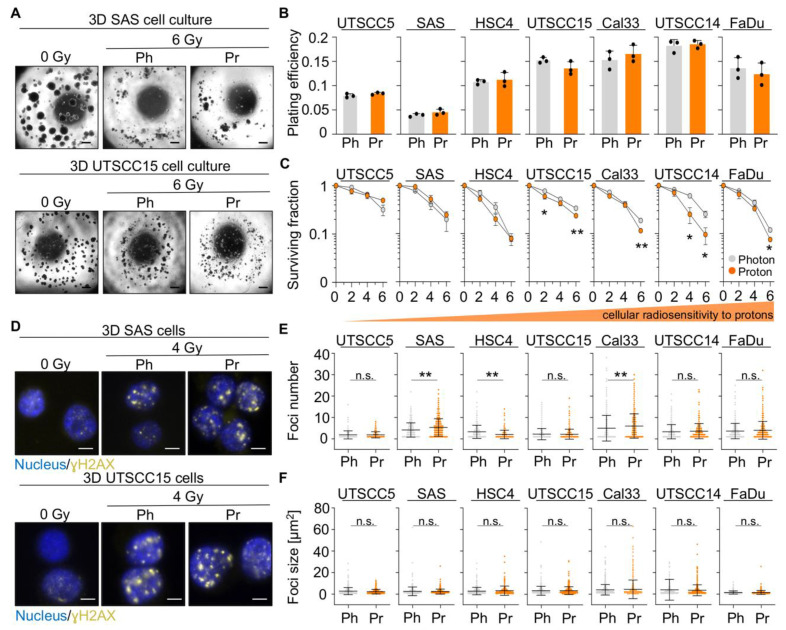
The intrinsic cellular radiosensitivity to photon and proton irradiation varies among 3D lrECM grown HNSCC cell cultures. (**A**) Representative microscopy images of unirradiated and irradiated 3D lrECM SAS and UTSCC15 cell colonies. Scale bar, 200 µm. (**B**) Plating efficiencies of unirradiated 3D lrECM HNSCC cell cultures. (**C**) Clonogenic radiation survival of indicated HNSCC cell lines upon photon or proton irradiation. (**D**) Representative immunofluorescence images of residual ɣH2AX foci at 24 h post 4-Gy irradiation (Scale bar, 20 µm; ɣH2AX in yellow and nuclei in blue). (**E**) Dot plots of residual foci numbers and (**F**) foci sizes 24 h post photon or proton exposure (4 Gy). At least 100 cells were analyzed per biological replicate. Results show mean ± SD (*n* = 3; two-sided *t*-test; * *p* < 0.05, ** *p* < 0.01). Ph, photon irradiation; Pr, proton irradiation; n.s., non-significant.

**Figure 2 cancers-13-01190-f002:**
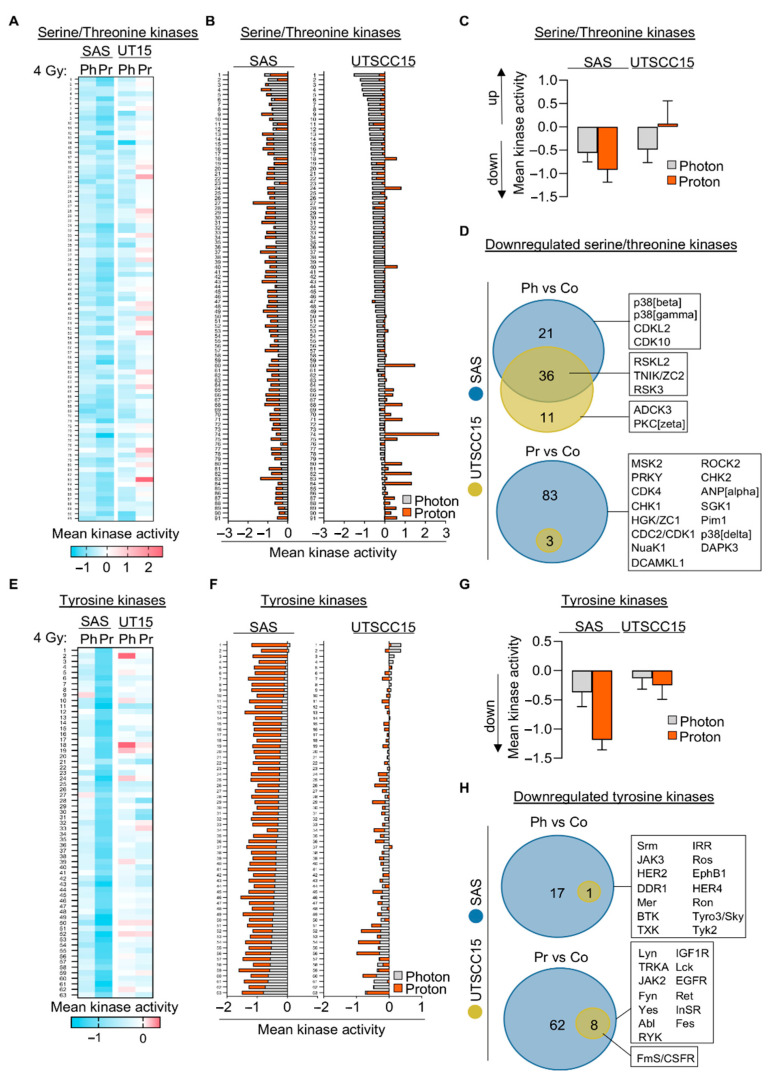
Photon- and proton irradiation induce differential kinome signatures. (**A**) Heatmap and (**B**) superimposed waterfall plot of 91 serine/threonine kinase activities (STK) 2 h after 4-Gy photon or proton irradiation in 3D SAS and UTSCC15 cell cultures normalized to unirradiated controls. (**C**) Mean activity of all investigated STK 2 h after 4-Gy photon or proton irradiation in 3D SAS and UTSCC15 cell cultures normalized to controls. (**D**) Venn diagrams of uniquely and jointly in photon- or proton-irradiated SAS and UTSCC15 cell cultures downregulated STK (mean kinase activity ≤−0.5). Boxes display significantly downregulated STK. (**E**) Heatmap and (**F**) superimposed waterfall blot of 63 phosphotyrosine kinase (PTK) activities at 2 h after 4-Gy photon or proton irradiation in 3D SAS and UTSCC15 cell cultures lines normalized to controls. (**G**) Mean activity of all investigated PTK at 2 h after 4-Gy photon or proton irradiation in 3D SAS and UTSCC15 cell cultures normalized to controls. (**H**) Venn diagrams of uniquely and jointly in photon- or proton-irradiated SAS and UTSCC15 cell cultures downregulated PTK (mean kinase activity ≤−0.5). Boxes display significantly downregulated PTK. Ph, photon irradiation; Pr, proton irradiation; vs, versus.

**Figure 3 cancers-13-01190-f003:**
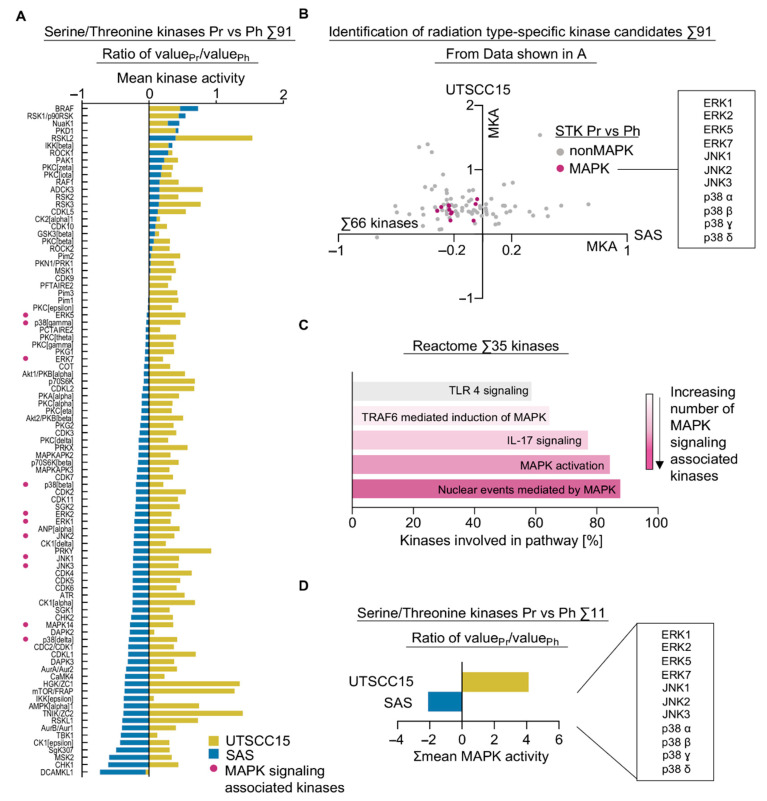
Cell line-dependent kinome alterations upon proton versus photon irradiation. (**A**) Comparative plotting of activity changes in 91 STK in 3D cell cultures 2 h after 4-Gy proton versus photon irradiation. (**B**) Comparative and differential visualization of STK mean kinase activities after 4-Gy proton versus photon irradiation. Sixty-six kinases in the left upper quadrant represent possible targets including the clustering of MAPK associated kinases. Purple colored dots represent mitogen-activated protein kinase (MAPK) signaling associated kinases; gray colored dots represent non-MAPK proteins; MKA, mean kinase activity. (**C**) Reactome based pathway analysis of 35 kinases of the left upper quadrant (mean kinase activity cut-off for SAS ≤ −0.2 and for UTSCC15 ≥ 0.2). (**D**) Summation of mean kinase activities of 11 selected kinases belonging to the MAPK pathways in both cell lines upon 4-Gy proton versus photon treatment. Ph, photon irradiation; Pr, proton irradiation; vs, versus.

**Figure 4 cancers-13-01190-f004:**
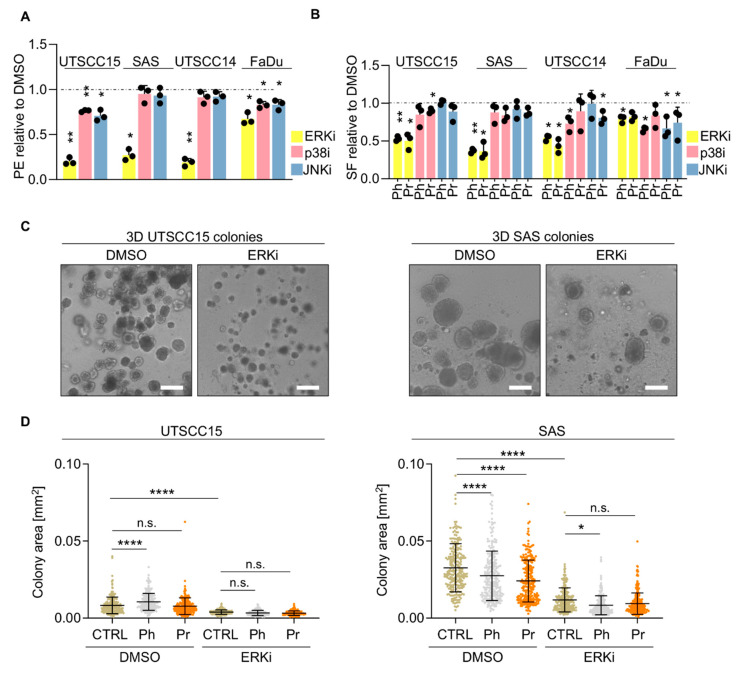
MAPK inhibitors mediate differential radiosensitizing potential in 3D HNSCC cell cultures exposed to photon or proton irradiation. (**A**) Cytotoxic effects of ERKi (Ulixertinib), p38i (Ralimetinib) and JNKi (SP600125) (all at a final concentration of 1 µM) in 3D cell cultures of indicated cell lines normalized to DMSO controls. (**B**) 3D clonogenic radiation survival of indicated HNSCC cell lines upon pretreatment with indicated inhibitors (all at a final concentration of 1 µM) normalized to DMSO controls. (**C**) Representative microscopy images of unirradiated 3D UTSCC15 and SAS cell colonies. Scale bar, 200 µm. (**D**) Colony sizes of 3D lrECM grown UTSCC15 and SAS colonies upon 4-Gy photon or proton irradiation in presence of the ERK inhibitor Ulixertinib. DMSO served as control. Results show mean ± SD (*n* = 3; two-sided *t*-test; * *p* < 0.05, ** *p* < 0.01; **** *p* < 0.0001). PE, Plating efficiency; SF, Surviving fraction; CTRL, control; Ph, photon irradiation; Pr, proton irradiation; n.s., non-significant.

**Figure 5 cancers-13-01190-f005:**
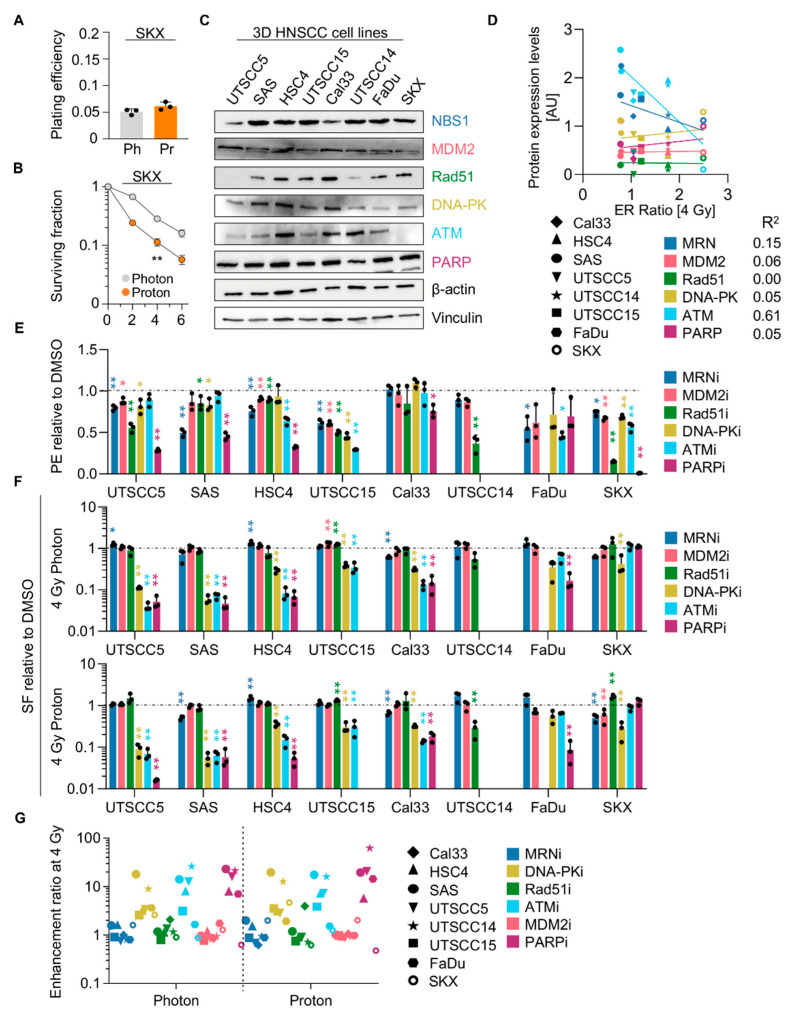
Specific DNA repair inhibitors mediate radiosensitization without discrimination between photon and proton irradiation. Plating efficiency (**A**) and clonogenic radiation survival (**B**) of 3D SKX cell cultures upon 2, 4, 6 Gy photon or proton irradiation. (**C**) Western blot analysis of indicated DNA repair proteins in indicated 3D HNSCC cell cultures. β-actin and Vinculin served as loading controls. (**D**) Correlation between enhancement ratios (ER; calculated by dividing the surviving fraction at 4-Gy proton irradiation by the surviving fraction at 4-Gy photon irradiation) and protein expression levels (see Appendix A). (**E**) Normalized plating efficiencies of 3D HNSCC cell cultures upon pretreatment with indicated DNA repair inhibitors (all at 10 µM). DMSO served as control. (**F**) Normalized clonogenic survival of 3D HNSCC cell cultures upon 4-Gy photon or proton irradiation in presence of DNA repair inhibitors (all at 10 µM) relative to DMSO controls. (**G**) Enhancement ratios for combined inhibitor/4-Gy irradiation. All results show mean ± SD (*n* = 3, two-sided *t*-test, * *p* < 0.05, ** *p* < 0.01). Ph, photon irradiation; Pr, proton irradiation; PE, Plating efficiency; SF, Surviving fraction.

## Data Availability

The data presented in this study are available on request from the corresponding author. The data are not publicly available due to our institutional guidelines.

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
