# Peer review of "Comparative Therapeutic Exploitability of Acute Adaptation Mechanisms to Photon and Proton Irradiation in 3D Head and Neck Squamous Cell Carcinoma Cell Cultures"

_cancers, 2021, doi:10.3390/cancers13061190_

Round 1

Reviewer 1 Report

Interesting and thorough paper on a large number of head and neck tumor cells (HNSCC) in vitro, albeit partially in 3 D cultures which is a better surrogate for in vivo but still a surrogate. So the results are valid in vitro but we don’t know in situ. Moreover they do not use primary cells but established lines with the intrinsic limitations. But all together a very worthwhile basic radiobiological comparison of proton vs proton radiation along some standard endpoints such as clonogenic survival and DNA double strand breaks and novel comparisons such the kinome profiles upon radiation. Interestingly the kinome profiles showed quantitative differences in many kinases such as the MAPK but the respective inhibitors showed no radiosensitization. In fact the radiation-induced activity changes of cytoplasmic kinases could not be transferred to sensitization of HNSCC cells to photon nor proton irradiation in the clonogenic assay at all.  Likewise, despite different kinome responses, inhibitors for ATM, DNA-PK and PARP did not discriminate between proton and photon irradiation but generally elicited a general radiosensitization.  So the authors found marginal cell line-specific differences in radiosensitivity in clonogenic survival and DSB repair, but no superiority of one radiation type over the other in cell cultures. The data look all solid, and robust, and valid; the analyzes are technically of high standard. I believe the data are also meaningful and important as a message in vitro and for the endpoints measured. The data certainly can fill a gap of knowledge in radiobiology.

But I suggest the authors could discuss at least in more detail the limitations of the study such as the non effect (no difference phot-prot) or non relevance statements may not be true for other endpoints measurable even in vitro, meaning that the conclusion is just not so general. In fact I think there is conflicting literature and also literature which suggests that the kinome responses can regulate the radiation response. Then the problem of the word "acute" - only a short time frame is covered naturally. Next and the obvious, the study is in vitro and cannot cover the response mechanism in vivo – which needs to be discussed in more detail.

Reviewer 2 Report

 In this manuscript, the authors hypothesized that rapid changes in phosphorylation state were important as an acute adaptation mechanism to photon or proton irradiation of head and neck cancer cells in three-dimensional culture, and compare the two cell lines by performing kinome analysis. In SAS cells, the activities of serine/threonine kinase and tyrosine kinase were greatly attenuated by photon or proton irradiation, whereas in UTSCC15 cells, the activities of some MAPK family kinases were increased by proton irradiation. Based on these results, the authors investigated the effects of MAPK inhibitors on radiosensitivity in several H&N cancer cell lines. Independently of the kinome experiment, the authors also investigated the effect of DNA repair inhibitors on radiosensitibity. Although these data may potentially contribute to the development of future research, I am afraid that the readers cannot understand what the authors really want to claim in this MS.

Major comments

(1) The most difficult point to understand in this study is why SAS and UTSCC15 were selected for the comparative analysis of kinome. As described in page 3, line 109-110, the response to proton irradiation is "relatively similar” among them. Why did the authors decide to compare them? Why didn't they compare cell lines with significantly different responses to proton irradiation, such as UTSCC5 and UTSCC14? In addition, in the comparative analysis of kinomes, the authors found that the response of MAPK family kinases was different between SAS and UTSCC15, and treated the cells with inhibitors of these kinases. I do not understand the expected effect of treatment with inhibitors of MAPK family kinases, whose activity alteration patterns after protein irradiation were different among the two cell lines that were "relatively similar" in their response to proton irradiation. In stead, it would be interesting to perform kinome analysis of UTSC5 and UTSC14 after photon and proton irradiation and show that the use of inhibitors of the kinases identified therein increases the radiosensitivity of UTSC5 and decreases that of UTSC14.

(2) Page 6, line 160-162: This is also observable in the TCGA HNSCC dataset where the mRNA expression levels of different MAPK kinase are positively or negatively correlated (Figure S2B).

I understand that the experiments showed that there is a difference in the variation of kinase activity of the MAPK family kinases in SAS and UTSC15 cells upon photon or proton irradiation, but what is the relevance of the TCGA dataset data?  A more detailed explanation is required.

(3) Page 8, line 205-209: Interesting in this context, ...for low versus highly expressed MAPK molecules (Figure S4B, D, F, H).

How does this data relate to the conclusions that the authors wish to present in this manuscript? It is easy to imagine that MAPK family kinases, which are related to various pathophysiologies including cell proliferation, can affect patient prognosis by unraveling the vast amount of past literature.

(4) In Figure 5, the authors show that DNA repair inhibitors enhance radiosensitivity, and that these effects do not differ between photons and protons. How does this relate to what the authors have shown in Figures 1 to 4? Also, how does this contribute to the "acute adaptation mechanism" that the authors want to clarify in this paper?

Minor comment

(1) In Figure 4, the authors examined whether MAPK inhibitors affected radiosensitivity; although the specificity of inhibitors of ERK and p38 is relatively high, SP600125, an inhibitor of JNK, seems not to be suitable for their purpose, because it also shows inhibitory activity against a wide variety of serine/threonine kinases and receptor-type tyrosine kinases such as TrkA. Moreover, since the target kinases have been identified in Figure 3, the authors should validate the effect by performing RNAi studies. The same thing can be said in Figure 5.

Reviewer 3 Report

This manuscript presents a comprehensive landscape on cellular event and possible differences between photon and proton irradiation, with implications on radio-sensitization strategy. Generally well-written with diligent and scientific presentation. Just a few minor issues are better to be commented.

1. 88, 89 ‘obvious changes in activation patterns of serine/threonine (STK) (Figure 2A and E) and phosphotyrosine kinases (PTK) (Figure 2B and F) in both cell lines ~’ should be (STK) (Figure 2A and B) and (PTK) (Figure 2E and F)

2. The limitation of this study, such as culture method-specific differences, lack of STED microscopy (as you mentioned), and lack of relevant in vivo or real-world data is better to be mentioned in more detail. 

Reviewer 4 Report

This manuscript clearly explored cell signaling of head and neck cancer cells exposed to protontherapy in comparison to photon therapy. Using kinome assays, authors showed that exposure to proton modulated the activity of members of the MAPK family in 2 tumor cell models in 3D culture. However, SAS and UTSCC15 cells show opposing activation patterns after protons. Furthermore, inhibition of different MAPK does not modulate tumor cell photon or proton radiosensitivities.  In the other hand, separated inhibition of most of the DNA repair pathways is increased radiosentivities.
The manuscript is well-written and the conclusion are dedicated to the proposed experiments. The authors explained clearly the limits of their results.

I consider the manuscript acceptable for publication.

Round 2

Reviewer 1 Report

all questions well addressed  

Reviewer 2 Report

The authors have entirely answered my questions and adequately explained their logic in the MS. I have no further comment on this MS. 

Reviewer 3 Report

Authors' revision looks good, diligently reflecting reviewer's previous comments.